# Antioxidant and Antibacterial Effect of Fruit Peel Powders in Chicken Patties

**DOI:** 10.3390/foods11030301

**Published:** 2022-01-23

**Authors:** Heba H.S. Abdel-Naeem, Hend Ali Elshebrawy, Kálmán Imre, Adriana Morar, Viorel Herman, Raul Pașcalău, Khalid Ibrahim Sallam

**Affiliations:** 1Department of Food Hygiene and Control, Faculty of Veterinary Medicine, Cairo University, Giza 12211, Egypt; 2Department of Food Hygiene and Control, Faculty of Veterinary Medicine, Mansoura University, Mansoura 35516, Egypt; hend_ali85@mans.edu.eg (H.A.E.); khalidsallam@mans.edu.eg (K.I.S.); 3Department of Animal Production and Veterinary Public Health, Faculty of Veterinary Medicine, Banat’s University of Agricultural Sciences and Veterinary Medicine “King Michael I of Romania”, 300645 Timişoara, Romania; adrianamo2001@yahoo.com; 4Department of Infectious Diseases and Preventive Medicine, Faculty of Veterinary Medicine, Banat’s University of Agricultural Sciences and Veterinary Medicine “King Michael I of Romania”, 300645 Timişoara, Romania; viorel.herman@fmvt.ro; 5Department of Agricultural Technologies, Faculty of Agriculture, Banat’s University of Agricultural Sciences and Veterinary Medicine, ”King Michael I of Romania”, 300645 Timisoara, Romania; raulpascalau@yahoo.com

**Keywords:** chicken patties, fruit peel powders, physicochemical, phenolics, flavonoids, DPPH%

## Abstract

Meat industries are eager to find natural low-cost additives for improving the health benefits and shelf life of meat products. The present study elucidated the effect of four different fruit peel powders, namely lemon, orange, grapefruit, and banana (1% each), on the oxidative stability, microbial quality, physicochemical properties, and sensory attributes of chicken patties during 3 months of storage at −18 °C. The total phenolics and flavonoids as well as the antioxidant activity of the fruit peel powders were analyzed. The lemon peel powder contained the highest bioactive substance (90.5 mg gallic acid/g total phenolics and 35 mg rutin/g total flavonoids) and had the highest free radical scavenging activity (90%). The fruit peel powders used, especially the banana peel powder, induced an increase in protein (22.18 g/100 g) and a decrease in fat (10.52 g/100 g) content. Furthermore, all the fruit peel powders exhibited significant antioxidant and antibacterial activities compared with the control samples. The sensory attributes were improved in all treated groups, especially in the lemon peel powder-treated patties. Consequently, the obtained results support the application of fruit peel powders, as natural sources of antioxidants with antibacterial effects, as health-promoting functional additives during the manufacturing of meat products.

## 1. Introduction

Chicken meat and its derivatives are one of the most widely consumed food products throughout the world. This may be due to the fact that this meat is not subjected to any cultural or religious limitations. Additionally, its consumption has been increasing rapidly due to its healthy and nutritional image. In this regard, the amount of proteins, essential amino acids, and vitamins are well presented along with a relatively low fat and cholesterol content [1]. However, their high moisture content with favorable pH and water activity along with their content of unsaturated fatty acids make such meat highly perishable and an ideal medium for bacterial growth and lipid oxidation, hence causing shelf life quality deterioration [2]. Furthermore, during the manufacturing of chicken meat products, the mincing process, incorporation of air, and presence of non-heme iron together with the generated heat cause disruption in the cell membranes of muscle and facilitate the interaction between unsaturated lipids and pro-oxidant substances, with a subsequent negative impact on the processed meat products from a sensory and health point of view.

The oxidative degradation could be successfully reduced by using synthetic antioxidants, including butylated hydroxytoluene (BHT) and butylated hydroxyanisole (BHA). Nevertheless, the use of these compounds is limited due to their toxicity and damaging effects at the DNA level [3]. Currently, increasing consumer awareness about the possible health implications of using synthetic additives has promoted consumers to establish prohibitive requirements for their foods, which should be nutritious, of high quality, and free from any chemical antioxidant or preservative due to the safety demands and the health hazards. This has encouraged meat industries to exploit plant derived additives in meat systems as an alternative to the use of synthetic ones. Therefore, fruits, vegetables, herbs, and other plant extracts or powders are commonly used as natural preservatives to improve the quality of meat products and extend their viability [4,5]. Fruit by-products have gained more attention due to their health benefits since they are considered an important source of bioactive substances, which are more concentrated in the peels compared to other parts of the fruit [6]. Such bioactive compounds can offer various health benefits owing to their antioxidant, antimicrobial, anticarcinogenic, antimutagenic, antiallergenic, and antiaging activities [7]. Moreover, they are responsible for the odor of aromatic plants; therefore, they can be used for the production of food flavor additives [7]. In this concern, fruit peels could be used in the manufacturing of meat products as natural additives, not only to improve shelf stability through delaying microbial growth and lipid oxidation, but also to produce low-cost and highly nutritious products with good organoleptic characteristics and physicochemical properties. Additionally, fruit peels are considered a source of environmental pollution; hence, their full utilization can help the industry and the authority to lower waste agribusiness and increase industrial profitability, besides adding value to the peel waste [8].

It has been previously demonstrated that some citrus peel extracts, such as lemon, orange, and grapefruit, as well as banana peel possess good antioxidant and antimicrobial properties due to their phenolic acid and flavonoid contents [9,10]. Furthermore, citrus-derived products are very rich in dietary fibers. Therefore, their incorporation into meat products could result in benefits for human health. It is noteworthy that most of the previous research has focused on using citrus fibers as a fat replacement to produce low-fat meat products [11], while the oxidative stability of meatballs using citrus peel extracts was studied by Nishad et al. [12]. Owing to the flavor considerations of fruit peel aromatic oily liquid extracts, the use of peel powders of citrus fruits, such as lemon, orange, and grapefruit, as well as banana during the processing of meat products is a good choice; nonetheless, studies about the application of fruit peel powders in the meat industry are very limited. The current study was undertaken to investigate the efficiency of lemon, orange, grapefruit, and banana peel powders in improving the microbial quality, antioxidant activity, and sensory characteristics of chicken patties. In addition, the shelf life-extending capacity of these fruit peel powders in chicken patties during cold preservation (−18 °C) for a period of 3 months was evaluated.

## 2. Materials and Methods

### 2.1. Preparation of Fruit Peel Powders

Fresh lemons, oranges, grapefruits, and bananas were acquired from a local retail market. After washing, the fruit peels were removed using a sharp kitchen knife in the case of the citrus fruits, and by hand in the case of the banana, while the edible portions were separated. The resulting peels were cut into small pieces using a sharp knife, dried in an air oven drier (Faithful, 101-3AB, China) at 50 °C for 48 h, and then minced to a fine powder with a kitchen grinder. The powders were passed through a 100 mesh sieve, packed in polyethylene bags, and subsequently stored at 20 °C until further use.

### 2.2. Ingredients Preparation

The deboned chicken meat (~15 kg of thigh and breast) provided by a poultry processing plant was trimmed of all visible fat and connective tissue and then stored in frozen conditions (−18 °C) for the next day. Common salt and starch were purchased from a local market in Giza, Egypt, while seasoning mix and sodium tripolyphosphate were purchased from Loba Chemie (Mumbai, India).

### 2.3. Product Formulation

A simple traditional formulation was used to prepare a base batter for the chicken patties as follows: 80% chicken breast and thigh (equal portions, 40% each), 7.5% plant oil, 1.8% sodium chloride, 0.3% sodium tripolyphosphate, 0.3% seasoning mix, 5% water, and 5% starch. From this base batter, four formulations were prepared by addition of 1% of each fruit peel powder (lemon, orange, grapefruit, and banana), besides a fifth formulation that was kept as a control with no fruit peel powders added.

### 2.4. Product Processing and Storage

The chicken patty formulations were processed at three independent occasions. For each occasion, the cooled chicken breast and thigh meat were ground together using a plate grinder (ø 4.5 mm; Seydelmann KG, Stuttgart, Germany). The minced chicken meat was mixed with salt, water, polyphosphates, oil, seasonings, and starch. The obtained mixture was divided into five groups comprising the first, second, third, and fourth groups that were treated with 1% of lemon, orange, grapefruit, and banana peel powders, respectively, and a 5th group that constituted the control, without any treatment. Subsequently, the mixture of each formulation was homogenized for 3 min by hand using sterile gloves, and was then formed into 1 cm thick discs weighing 75 g using a manual patty former obtained from Fac Affettatrici (Lombardia, Italy). The chicken patties were packaged in plastic films and kept at −40 °C for 30 min. Next, the samples were placed in plastic containers and stored under frozen conditions (−18 °C) for three months. The samples were enrolled in triplicates from each formulation for analysis on the second day (time zero) from storage and then monthly.

### 2.5. Examinations

#### 2.5.1. Total Phenolic Content Analysis in Fruit Peel Powders

The total phenolic content of the analyzed samples were measured using the Folin–Ciocalteau reagent, as has been previously described by Singleton et al. [13]. In order to obtain the sample extract, 2 g of fruit peel powder were mixed in a methanol aqueous solution (80%) and then centrifuged at 10,000 rpm for 15 min, using a refrigerated centrifuge (Jouan CR 4–22, Riverview, FL, USA). The resulting supernatant was collected and the remaining sediment was re-extracted afterwards with methanol (80%). The whole supernatant was placed into evaporating dishes and allowed to dry at room temperature, and the resulting residue was dissolved in 5 mL of distilled water (DW). One milliliter of fruit peel extract, 5 mL of Folin–Ciocalteu solution, and 60 mL of DW were added into a 100 mL volumetric flask. After an exposure time of 4 min, 15 mL of sodium carbonate (20%) were added. Next, the total volume of the mixture was adjusted with DW to 100 mL, followed by incubation for 2 h in a dark place. Finally, the absorbance of the resulting blue color was quantified at 760 nm against gallic acid (GA) as a standard using a spectrophotometer (Unico 1200, Dayton, NJ, USA). The obtained results were expressed as mg GA/g.

#### 2.5.2. Total Flavonoid Content Analysis in Fruit Peel Powders

The determination of the total flavonoid content of the fruit peel powders was achieved using the aluminum chloride colorimetric method, according to the protocol described by Zhishen et al. [14]. One milliliter of the sample extract was introduced into a test tube containing 4 mL of DW and 0.3 mL of sodium nitrite solution (5%). After an exposure time of 6 min, 2 mL of NaOH (1 mol/L) and 0.3 mL of aluminum chloride (10%) solution were added, and the mixture was allowed to stand for an additional 5 min. The mixture was completed and mixed with 10 mL DW. The absorbance of the resulting color was measured against a blank at 510 nm using a spectrophotometer (Unico 1200, Dayton, NJ, USA) with rutin as standard (1 mg/mL). The obtained results were expressed as mg rutin/g.

#### 2.5.3. Antioxidant Activity Measurement of Fruit Peel Powders Using DPPH Assay

The determination of the antioxidant activity of the fruit peel powder was carried out according to the procedure described by Brand-Williams et al. [15] based on the measurement of the free radical scavenging ability of the tested powders towards the stable 1,1-diphenyl-2-picrylhydrazyl (DPPH) radical. Briefly, 100 µL of the sample extract was homogenized with 3.9 mL of DPPH solution (0.0634 mM) in methanol (95%) for up to 10 s, followed by incubation at room temperature for 30 min in a dark place. The solution absorbance was quantified at 515 nm against a blank of methanol without DPPH using a spectrophotometer (Unico 1200, Dayton, NJ, USA). The DPPH radical scavenging activity, expressed as a percentage, was computed using the following equation:(1)$$\mathrm{DPPH}\;\left(\%\right)=\frac{\left(\mathrm{AD}\;-\;\mathrm{AS}\right)}{\mathrm{AD}}\;\times \;100$$
where AD is the absorbance of the methanolic solution of DPPH (without sample) and AS is the sample absorbance.

#### 2.5.4. Proximate Compositional Analysis of Chicken Patties

The compositional analysis of chicken patties from the different groups was carried out following the methods recommended by the Association of Official Analytical Chemists (AOAC) [16]. Thus, within the moisture content (g/100 g) determinations, the drying of 10 g of sample was carried out at 100 °C until a constant weight was obtained. The samples’ protein content (g/100 g) was analyzed using the Kjeldahl method, and the final crude protein content was obtained, after the conversion of the nitrogen content, using a factor of 6.25. The fat content (g/100 g) was estimated using a Soxhlet apparatus, while the ash content (g/100 g) was obtained using a muffle furnace at 500 °C for 5 h.

#### 2.5.5. Bacteriological Examination of Chicken Patties

The bacteriological examination of the chicken patties from different groups was performed on the second day after processing, and then monthly during frozen storage. The enumeration of the aerobic plate count (APC) was achieved after a 48 h incubation of the inoculated standard plate count agar plates (CM 463; Oxoid Ltd., Basingstoke, UK) at 32 °C; the presence of psychrotrophic bacteria was determined after a 7 day incubation of the standard plate count agar plates at 7 °C. Presumptive *Staphylococcus aureus* colonies were counted on double sets of the inoculated Baird-Parker agar plates (CM 145; Oxoid Ltd., Basingstoke, UK) after an incubation period of 48 h at 37 °C. The Enterobacteriaceae counts were determined using violet red bile glucose agar (CM 1082; Oxoid Ltd., Basingstoke, UK) after an incubation period of 24 h at 37 °C.

#### 2.5.6. Deterioration Criteria Measurement of the Chicken Patties

The pH, TVBN, and TBA values of the chicken patties from different groups were measured after processing and each month during frozen storage. A digital pH meter (SensoDirect 150, Lovibond, Sarasota, FL, USA) was used for the pH value measurements after the homogenization of 5 g of sample with 20 mL DW for 10 s and the calibration of the pH meter using pH 7.0 and 4.0 buffer solutions [17]. The macro-Kjeldahl distillation method was used for the total volatile base nitrogen (TVBN, mg/100 g) estimation [18]. The measurement of the thiobarbituric acid value (TBA, mg malondialdehyde/kg) was carried out according to the procedure described by Du and Ahn [19].

#### 2.5.7. Color Evaluation of Chicken Patties

The color of chicken patties from different groups was measured after processing and monthly during frozen storage using a Chroma Meter (CR 410, Konica Minolta, Marunouchi, Japan) with an illuminant D65, 11 mm aperture size, and 10° observer, according to the method described by Shin et al. [20]. The meter was calibrated against a white CR410 calibration plate (Y = 94.40, x = 0.3159, y = 0.3325). For each sample, three readings were taken and the mean value was calculated and expressed as CIE lightness (L*), redness (a*), and yellowness (b*).

#### 2.5.8. Sensory Analysis of Chicken Patties

The sensory analysis of the chicken patties was carried out after processing and each month during storage (−18 °C), according to the recommended guidelines from the American Meat Science Association (AMSA) [21]. For sensory evaluation, odd numbers of well-trained panelists from the Food Hygiene and Control Department were selected on the basis of their experience in the assessment of meat products. Chicken patties from each formulation were placed on a clean aluminum foil plate and cooked in an electrical draught oven (180 °C) (Faithful, 101-3AB, China) at a central temperature of 75 °C. Three replicates from the four formulations, along with the controls, were served in a randomized order and evaluated by the panelists. Numerical values between 1 (meaning extremely unacceptable) and 9 (meaning extremely acceptable) were assigned by each expert concerning the appearance, flavor intensity, juiciness, tenderness, and overall acceptability of the cooked chicken patties.

### 2.6. Statistical Analysis

A triplicate measurement was applied for each of the tested samples. The statistical analysis of the obtained data was carried out using the IBM SPSS statistics software for Windows, version 27.0 (IBM Corp., Armonk, NY, USA). A one-way analysis of variance (ANOVA) was used for the determination of differences between the means of the values of the recorded measurements. In addition, multiple comparisons of the means were achieved using the post hoc (least square difference, LSD) test. A *p* value ≤ 0.05 was considered significant.

## 3. Results and Discussion

### 3.1. Total Phenolics, Total Flavonoids, and DPPH% of Fruit Peel Powders

The analysis of the four different fruit peel powders prepared in the present study revealed that their phenolic and total flavonoid contents were significantly (*p* < 0.05) different, and were greatest in the lemon peel powder, followed by the orange peel powder, grapefruit peel powder, and banana peel powder (Figure 1).

Correspondingly, DPPH% activity also followed the same pattern as the results of the total phenolic and total flavonoid contents, where the highest antioxidant activity was observed in lemon peel powder and the lowest effect was found in banana peel powder (Figure 1).

The results obtained are in accordance with those of Gorinstein et al. [22], who reported that the total polyphenol content in lemon peels (190 mg chlorogenic acid/100 g) were significantly higher than those in orange (179 mg chlorogenic acid/100 g) and grapefruit (155 mg chlorogenic acid/100 g) peels, and they showed that lemon peels exhibited a higher antioxidant activity compared to orange and grapefruit peels. In another study, Abd El-Khalek and Zahran [23] found that orange peel powder had a higher DPPH% than grapefruit peel powder. The total phenolic and flavonoid contents of the banana peel were 17.89 mg GA/g and 21.04 mg quercetin/g, respectively, while the DPPH % was 24.18% and 45.76% at concentrations of 5 μg/mL and 50 μg/mL, respectively [10].

Contrary to our results, Sir Elkhatim et al. [24] found that the total phenolic content (77.3 mg GA/g) in grapefruit peels was significantly higher than that in lemon (49.8 mg GA/g) or orange peels (35.6 mg GA/g), and they indicated that the total flavonoid content of orange peels (80.8 mg catechin/g) and grapefruit peels (83.3 mg catechin/g) were significantly higher than those of lemon peels (59.9 mg catechin/g); although, they reported a significantly higher DDPH% in grapefruit peel (76.4%) and lemon peel (73.2%) than that in orange peels (70.5%). On the other hand, Czech et al. [6] reported that the total phenolic content of orange peel (312.2 mg GA/100 g) was significantly higher than that in lemon peel (251.1 mg GA/100 g) or that in grapefruit peel (208.7 mg GA/100 g). The variations between the different studies may be related to the differences in the nature and origin of the plant species, the environmental conditions, and the extraction solvent used.

The phenolic compounds in the fruit peel powders possessed a strong antioxidant and antimicrobial activity. The recorded positive correlation between the total phenolics and antioxidant activity was not surprising since their contents are considered to be main contributors to the antioxidant capacities of the fruit residues [24]. In this concern, the higher total polyphenolic content will, at the same time, increase the antioxidant capacity [22]. The antioxidant activity of phenolic compounds may be elucidated by their metal chelating properties and free radical scavenging activity [6]. The aromatic ring of phenolic compounds has a hydroxyl group that enables them to scavenge peroxyl radicals and form a stable end product that will prevent the further oxidation of lipids. Nonetheless, the antimicrobial properties of phenolic compounds may be related to their capacity to destroy the cell wall, associated with the disruption of the cytoplasmic membrane, which results in the leakage of cellular compounds and the alteration of fatty acid and phospholipid constituents, as well as interferes with DNA and RNA synthesis [25]. Tannins are responsible for the antioxidant properties of citrus fruits [6]. However, the antimicrobial activity of lemon peels is related to their tetrazene and coumarin contents [26].

### 3.2. Proximate Composition Analysis of the Different Fruit Peel Powder-Treated Chicken Patties

The compositional analysis of the chicken patties treated with different fruit peel powders is presented in Table 1. Our findings revealed that the addition of 1% of lemon, orange, grapefruit, or banana peel powders into the formulation of the chicken patties induced insignificant (*p* > 0.05) changes in the moisture and ash content, as compared with the control samples. Nonetheless, the protein content was significantly (*p* < 0.05) higher in banana peel powder-treated chicken patties, along with an insignificant (*p* > 0.05) increase in lemon, orange, or grapefruit peel powder-treated samples. On the other hand, a significant (*p* < 0.05) reduction in the fat content was noticed in all chicken patties treated with the different fruit peel powders, especially in those treated with the powder of banana peel.

Our findings are in accordance with those of Mahmoud et al. [27], who reported that the addition of 2.5% orange peel powder in beef burgers resulted in a significant reduction in fat content and insignificant modifications in protein and ash content, compared with samples from the control group.

In another study, Chappalwar et al. [28] found that the addition of 1% albedo lemon peel powder into low-fat chicken patties induced a statistically significant increase in moisture and decrease in fat content, along with insignificant changes in protein and ash content. Nevertheless, Haque et al. [29] recorded that the treatment of beef muscle with 0.2% orange peel extract caused a statistically significant decrease in fat and an increase in protein content. The higher protein content of banana peel-treated chicken patties among the other treatments, in the current investigation, may be related to the higher protein content in the peel. This observation was also reported by Wachirasiri et al. [30], who indicated that the protein content of banana peels was greater than that of orange and lemon peels. Likewise, Romelle et al. [31] obtained a higher protein content (10.44 vs. 9.73 g/100 g) and a slightly lower fat content (8.40 vs. 8.70 g/100 g) for banana peel in comparison to orange peel, respectively. Accordingly, the addition of lemon, orange, grapefruit, or banana peel powders into the chicken patties increased the nutritive values of such patties. These findings will help meat processors, especially in the lower-income countries, to produce meat products with high nutritive value and low prices.

### 3.3. Bacteriological Examination of the Different Fruit Peel Powder-Treated Chicken Patties

The results obtained revealed that the APC, psychrotroph, *S. aureus*, and Enterobacteriaceae counts were significantly (*p* < 0.05) lower in all fruit peel powder-treated chicken patties when compared with their counterpart untreated samples at the beginning of examination, on the second day of processing, and throughout the 3-month storage period at −18 °C (Table 2). This effect was more pronounced in the lemon and orange peel powder-treated chicken patties, followed by grapefruit peel powder-treated samples, while the least pronounced effect was recorded in the banana peel powder-treated samples. Interestingly, the psychrotroph, *S. aureus*, and Enterobacteriaceae counts were situated under the detectable limit (<2 log_10_ CFU/g) in lemon, orange, and grapefruit peel powder-treated samples at time zero of the examination and during the entire frozen storage period (Table 2). Nonetheless, *S. aureus* and Enterobacteriaceae counts were situated under the detectable limit (<2 log_10_ CFU/g) in banana peel powder-treated samples during the first month of frozen (−18 °C) storage, yet their recovery was observed during the examinations at the second and third months of storage (Table 2). It is noteworthy that all the examined bacterial counts were significantly (*p* < 0.05) increased during the storage time, especially in the control untreated samples, while such an increase was slow and quite less obvious in the fruit peel powder-treated samples (Table 2).

The obtained findings are in agreement with those reported by Ibrahim et al. [32], who showed that the addition of orange or lemon peel powders (1 or 2%) to beef patties induced a significant reduction in the APC during chilled storage at 4 °C for 15 days when compared with the control sample, and they indicated that the peel powders caused sudden lethal effects for microorganisms.

Likewise, a significant reduction in the APC in ground meat samples treated with orange and grapefruit peel powders with an extension of their shelf life to 21 days and 14 days, respectively, as compared with control samples that were microbiologically rejected on day 7 of storage at 4 °C was observed by Abd El-Khalek and Zahran [23]. This indicates that orange peel powder possesses a higher antimicrobial activity than grapefruit peel powder, and confirmed the data obtained in this study.

The bacterial counts in lemon, orange, and grapefruit peel powder-treated samples were lower than those of the control untreated sample, which may be due to their phenolic acid content (Figure 1). These phenolic compounds are natural constituents of such peels and exhibit a potent antibacterial activity by decreasing the internal pH value of microbial cells through the ionization of acid molecules, as well as by altering cell membrane permeability with the consequent disruption of substrate transport [29]. Additionally, the antimicrobial activity of citrus fruits can be explained by their ability to lower water activity inside the products. The antibacterial activity of banana peel has been observed previously by Mokbel and Hashinaga [33]. Similarly, the marination of beef with banana peel extracts (1%, 3%, and 5%) resulted in a significant reduction in the APC as well as *S. aureus* and Enterobacteriaceae counts in the marinated samples [34]. Moreover, Chabuck et al. [35] observed that fresh yellow banana peel aqueous extracts possessed good antibacterial activity against Gram-negative and Gram-positive bacteria.

### 3.4. Deterioration Criteria of the Different Fruit Peel Powder-Treated Chicken Patties

The registered pH, TVBN, and TBA values of the chicken patties with the incorporation of different fruit peel powders during frozen (−18 °C) storage for a period of 3 months are illustrated in Table 3. The pH value decreased significantly (*p* < 0.05) in the samples treated with lemon, orange, and grapefruit peel powders, and insignificantly (*p >* 0.05) in the samples treated with banana peel powder during frozen storage, as compared with the samples from the control group. However, the TBA and TVBN values were significantly (*p* < 0.05) decreased during storage in all the fruit peel powder-treated samples. Moreover, with the increase in storage time, all deterioration criteria were significantly (*p* < 0.05) increased in control samples and insignificantly (*p >* 0.05) increased in all the fruit peel powder-treated samples. Furthermore, among all treatment groups, the lemon peel powder-treated samples exhibited the lowest pH, TBA, and TVBN values, followed by the orange and grapefruit peel powder-treated samples, and lastly the banana peel powder-treated samples.

Our results are in harmony with those of Mahmoud et al. [27], who concluded that the treatment of beef burgers with different concentrations of orange peel powder (2.5%, 5%, 7.5%, and 10.0%) significantly decreased the pH, TBA, and TVBN values, as compared with the controls. Likewise, Ibrahim et al. [32] found that the addition of orange or lemon peel powders (1% and 2%) during the formulation of ground beef patties induced a significant reduction in pH, TBA, and TVBN values throughout storage at 4 °C for 15 days. Additionally, Sayari et al. [36] reported that the TBA values of grapefruit peel extract-treated turkey sausage were significantly lower than those of the control samples during storage at 4 °C for 13 days. Similarly, grapefruit peel extracts (1%) have been shown to retard the formation of malondialdehydes and lipid peroxides as well as the protein oxidation of goat meatballs during frozen (−18 °C) storage for 6 months [12]. In the same regard, Boraha et al. [37] observed that citrus and banana peel extracts were effective in maintaining low TBA values in chicken meatballs during refrigerated storage (4 °C) for 15 days, as compared to the controls.

The reduction in pH values of citrus fruit peel-treated chicken patties may be related to their higher organic acid content, including citric, oxalic, malic, and malonic acids [38]. Additionally, the lower TBA values of fruit peel powder-treated chicken patties indicate the antioxidant activity of such fruit peels.

The antioxidant activity of fruit peels has been attributed to their vitamin C, vitamin E, flavonoid, phenolic, and carotenoid contents [6]. Phenolic compounds have a free radical scavenging mechanism and have the ability to chelate metals with subsequent antioxidant potency. It is noteworthy that the higher antioxidant activity of lemon peel powder-treated samples, compared with the other groups, may be due to their higher total polyphenol content (Figure 1). Such an observation was also confirmed by Gorinstein et al. [22], who reported that the content of total polyphenols in lemons peels was significantly higher than that found in grapefruit and orange peels. The non-significant increase in pH, TBA, and TVBN values in all the fruit peel powder-treated samples during storage may be related to the inhibitory effect of the bioactive compounds in fruit peel powders against microbial growth and internal enzyme activities. In contrast, the significant increase in the values of the control untreated samples during storage owes to the development of spoilage bacteria.

### 3.5. Color Evaluation of the Different Fruit Peel Powder-Treated Chicken Patties

The results of color values revealed a significant (*p* < 0.05) decrease in the L* value in all fruit peel powder-treated chicken patties, especially those treated with banana peel powder, when compared with control untreated patties at time zero of examination and throughout the storage period (Table 4). Furthermore, the treatment of chicken patties with lemon, orange, and grapefruit peel powders induced significant (*p* < 0.05) increases in a* and b* values, as compared with the control values (Table 4). However, the treatment of chicken patties with banana peel powder resulted in a non-significant (*p >* 0.05) increase in a* values and a significant (*p* < 0.05) decrease in b* values during the entire storage period (Table 4). Additionally, L* values were significantly (*p* < 0.05) increased, and the reverse pattern occurred for a*and b* values with the increase in the storage period; this effect was more pronounced in the control sample and less noticeable in all the fruit peel powder-treated samples (Table 4).

The present results are in concordance with those demonstrated by Fernández-López et al. [39], who found that the incorporation of 5% of lemon or orange extracts into beef meatballs significantly decreased L* values and significantly increased a* and b* values during chilled (8 °C) storage for 12 days. Moreover, Nishad et al. [12] observed that the incorporation of grapefruit peel extract (1%) into goat meatballs significantly decreased L* values and significantly increased a*and b* values during frozen (−18 °C) storage for a period of 6 months. On the contrary, Abd El-Khalek and Zahran [17] demonstrated that the addition of orange and grapefruit peel powders into beef ground meat resulted in significant increases in L* values, non-significant decreases in a* values, and non-significant increases in b* values during refrigerated storage (4 °C) for 21 days. Additionally, Chappalwar et al. [28] noticed that the incorporation of lemon albedo powder into chicken meat patties generated a significant increase in L* value and had an insignificant effect on a* and b* values. Among all treatments, the lower L*, a*, and b* values were observed in the chicken patties treated with banana peel powder (Table 4). This finding was in agreement with that reported by Wachirasiri et al. [30], who demonstrated that the incorporation of banana peel powder within the food system could affect the color of the final product.

The marked increases in the L* values of the control samples during the storage period may be related to the scattered light reflections of the oxidized lipids. Furthermore, the significant decreases in the a* values of the control samples during the storage period may be attributed to the formation of metmyoglobin after myoglobin oxidation. In contrast, the significantly lower L* and higher a* values of lemon, orange, and grapefruit peel powder-treated samples, as compared with the control samples, are related to the protective and antioxidant effect of these fruit peel powders. Likewise, Fernández-López et al. [39] explained the significant decrease in the L* value of lemon and orange extract-treated beef meatballs by the hygroscopic characteristics of such materials, since these extracts were prepared from dry powders and had the ability to absorb free water and increase water retention within the product. In contrast, Abd El-Khalek and Zahran [23] attributed the higher L* values of orange and grapefruit peel powder-treated ground meat samples to the higher fiber content in such peels, which is composed of macromolecules and can rehydrate and remain outside the meat matrix. Furthermore, the significant increase in the b* value of lemon and orange extract-treated beef meatballs may have been due to the carotene pigment content in the citrus peels [39].

### 3.6. Sensory Analysis of the Different Fruit Peel Powder-Treated Chicken Patties

The addition of lemon, orange, and grapefruit peel powders into the formulations of chicken patties generated significant (*p* < 0.05) increases in appearance, flavor, tenderness, and juiciness scores, compared with the control samples (Figure 2 and Figure 3).

Likewise, the chicken patties treated with banana peel powder exhibited significant (*p* < 0.05) increases in flavor, tenderness, and juiciness scores, when compared with the control patties; however, they demonstrated an insignificant (*p* > 0.05) increase in appearance score (Figure 2 and Figure 3).

The overall acceptability score of samples treated with lemon, orange, and grapefruit peel powders was significantly (*p* < 0.05) increased at time zero of examination, and throughout the storage period, while in banana peel powder-treated samples it showed a significant (*p* < 0.05) increase only at the second and third months of the storage (Figure 4). Interestingly, among all treatments, the higher sensory scores were observed in lemon peel powder-treated samples, while the lower sensory scores were noticed in samples treated with banana peel powder (Figure 2, Figure 3 and Figure 4).

Our results are in harmony with those published by Ibrahim et al. [32], who found that the incorporation of orange or lemon peel powders (1% and 2%) into ground beef patties improved their sensory scores (appearance, color, odor, taste, and tenderness) during storage at refrigeration (4 °C), for 15 days. In addition, they indicated that the improvement in the sensory attributes was superior in lemon peel powder-treated samples over that in orange peel powder-treated samples, and it was better in the lemon concentration of 2% than in 1%.

Similarly, Chappalwar et al. [28] obtained higher sensory scores for 1% albedo lemon peel powder-treated chicken patties than those in the controls, and they revealed a significant decrease in the sensory scores with the increase in lemon peel added (2% and 3%), which they suggested was due to the increase in the yellowness values and pungent flavor, as well as the slightly bitter taste, as a result of the concentrations of limonin and naringin compounds inside the fruit during its development [40]. Likewise, the addition of 1% grapefruit peel extract into the formulation of goat meatballs significantly improved the color, odor, taste, and overall acceptability scores, especially in the third and sixth months of frozen storage [12].

Contrary to the results reported in this study, Mahmoud et al. [27] demonstrated that the treatment of beef burgers with orange peel powder (2.5%, 5%, 7.5%, and 10%) resulted in significant decreases of the sensory scores, as compared to control samples; meanwhile, among all treatments the highest scores were recorded in 5% orange peel powder-treated samples. In addition, Haque et al. [29] reported that there were non-significant improvements in the color, flavor, tenderness, juiciness, and overall acceptability scores of 0.2% orange peel extract-treated beef muscle, as compared to the control samples.

The improvement of the flavor of citrus fruit (lemon, orange, and grapefruit) peel powder-treated chicken patties was attributed to the presence of aromatic compounds, which are produced as metabolites during the usage of organic acids [41] or to their higher flavonoid compound content [7]. However, the higher tenderness score of lemon, orange, and grapefruit peel powder-treated samples than that in the control may be due to their lower pH values, which result in tenderization of the protein. This observation was also reported by Ibrahim et al. [32]. Moreover, Chappalwar et al. [28] found that the addition of 1% albedo lemon peel powder into chicken patties improved the tenderness, due to their higher moisture content, resulting in the improvement of the softness and juiciness of the tested product.

## 4. Conclusions

Healthy meat products without any synthetic additives are a main target for the consumer. Moreover, the prudent use of wasted fruit by-products assists in environmental protection and increases industrial profitability. Therefore, our study concluded that the incorporation of orange, lemon, grapefruit, or banana peel powders (1%) into chicken patties could improve their microbial quality, oxidative stability, and sensory attributes, and such improvements were more pronounced in the lemon peel powder-treated samples. Moreover, among the four fruit peel powders added, the total phenolic and flavonoid contents, together with the DPPH% activity, were significantly higher in the lemon peel powder. Additionally, these fruit peel powders may provide benefits for both meat processors and consumers by increasing the nutritional value of the product and lowering the formulation cost. Based on the promising results obtained in the current investigation, future studies focusing on the examination of the effect of frozen storage on phenicol and flavonoid contents as well as the antioxidant activity of other fruit peel powders containing bioactive compounds in chicken meat patties or in other meat-derived products are recommended.

## Figures and Tables

**Figure 1 foods-11-00301-f001:**
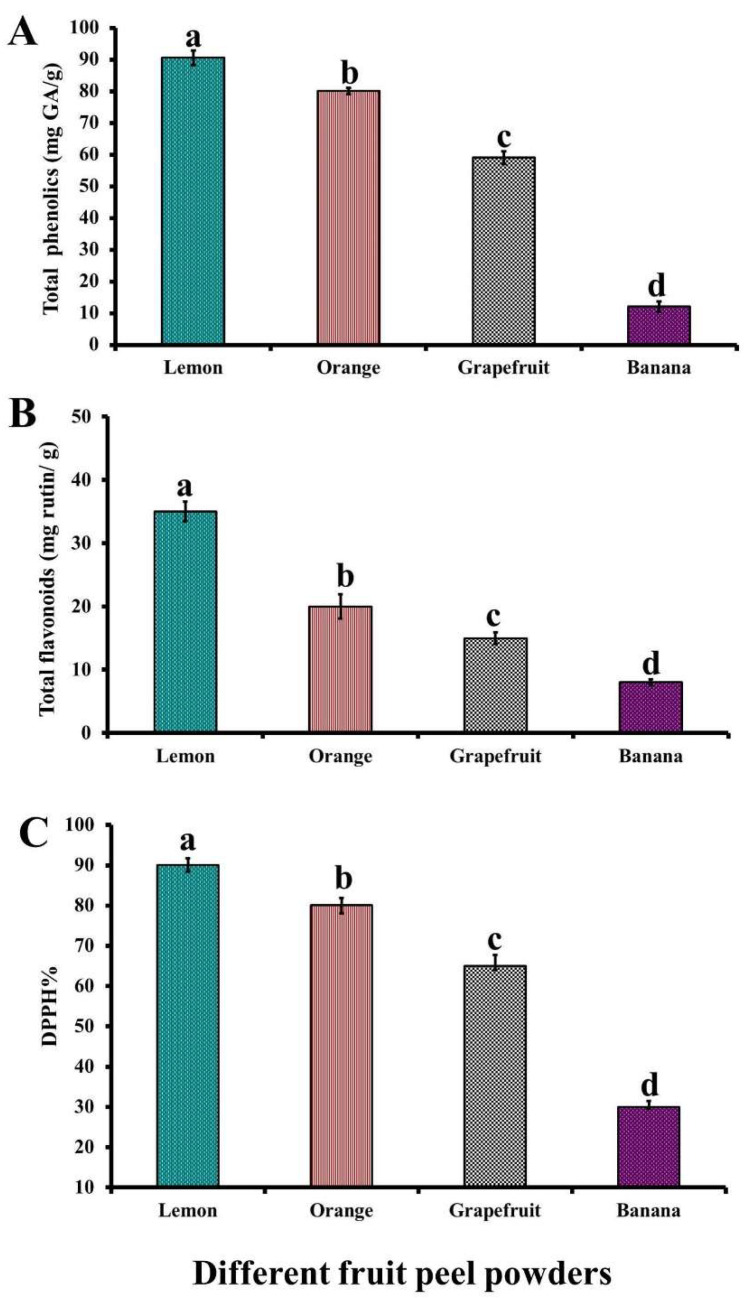
Total phenolic (**A**) and flavonoid (**B**) contents, and DPPH% (**C**) ac-tivity of different fruit peel powders. Columns marked by different letters showed significant differences (*p* < 0.05).

**Figure 2 foods-11-00301-f002:**
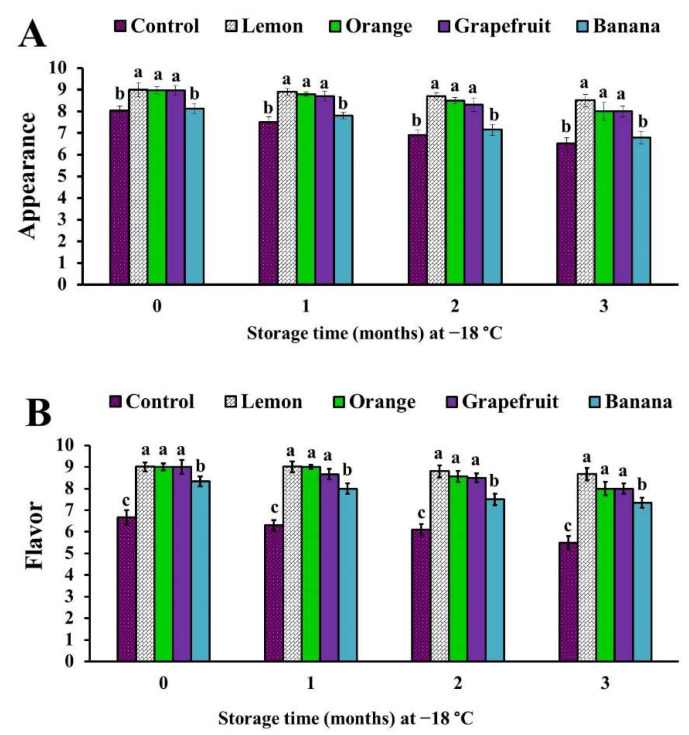
Appearance (**A**) and flavor (**B**) scores of chicken patties treated with different fruit peel powders. Columns marked by different letters indicate significant differences (*p* < 0.05).

**Figure 3 foods-11-00301-f003:**
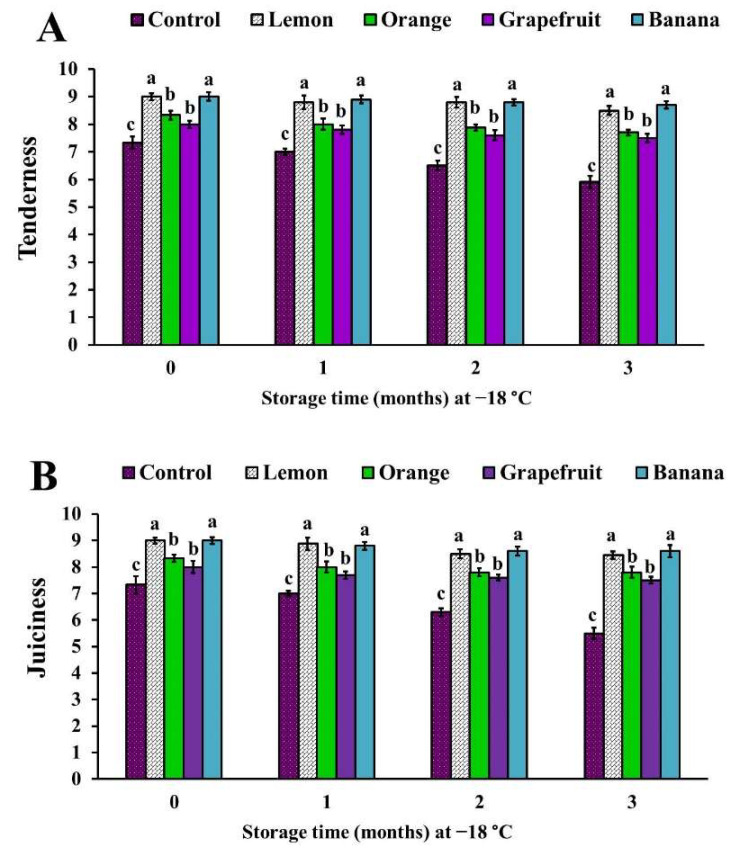
Tenderness (**A**) and juiciness (**B**) scores of chicken patties treated with different fruit peel powders. Columns marked by different letters indicate significant differences (*p* < 0.05).

**Figure 4 foods-11-00301-f004:**
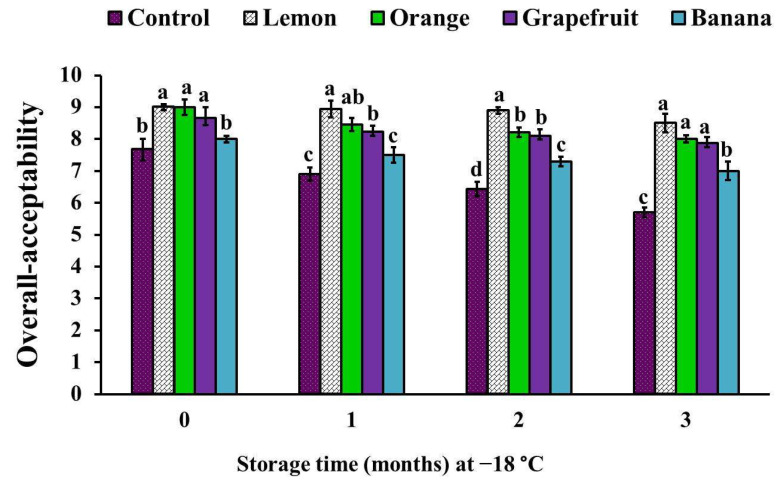
Overall acceptability scores of chicken patties with the incorporation of different fruit peel powders. Columns marked by different letters means significant differences (*p* < 0.05).

**Table 1 foods-11-00301-t001:** Results of the compositional analysis of chicken patties treated with different fruit peel powders at a concentration of 1%.

Proximate Composition	Treatments				
	Control	Lemon Peel Powder	Orange Peel Powder	Grapefruit Peel Powder	Banana Peel Powder
Moisture (g/100 g)	63.62 ^a^ ± 1.01	64.04 ^a^ ± 0.93	64.37 ^a^ ± 0.27	64.33 ^a^ ± 0.08	64.95 ^a^ ± 0.52
Protein (g/100 g)	18.71 ^b^ ± 0.71	19.17 ^b^ ± 1.01	19.89 ^b^ ± 1.02	19.80 ^b^ ± 0.95	22.18 ^a^ ± 0.38
Fat (g/100 g)	15.55 ^a^ ± 0.25	14.53 ^b^ ± 0.26	13.45 ^b^ ± 0.60	13.57 ^b^ ± 0.28	10.52 ^c^ ± 0.21
Ash (g/100 g)	2.09 ^a^ ± 0.15	2.24 ^a^ ± 0.09	2.28 ^a^ ± 0.12	2.29 ^a^ ± 0.10	2.34 ^a^ ± 0.11

^a–c^ Mean values with a different superscript letter within the same row are significantly different (*p* < 0.05). The presented values constitute the mean of triplicate measurements ± standard error (SE).

**Table 2 foods-11-00301-t002:** Microbial counts (log_10_ CFU/g) of chicken patties with the incorporation of different fruit peel powders (1%) during frozen (−18 °C) storage for 3 months.

Microbial Counts	Treatments	Storage Period (Months)			
		Time Zero	First Month	Second Month	Third Month
APC					
	Control	4.51 ^a,w^ ± 0.01	4.79 ^a,w^ ± 0.02	5.60 ^a,x^ ± 0.03	6.60 ^a,y^ ± 0.17
	Lemon peel powder	2.21 ^d,w^ ± 0.05	2.48 ^d,w^ ± 0.07	2.70 ^d,w^ ± 0.05	2.80 ^d,w^ ± 0.02
	Orange peel powder	2.34 ^d,w^ ± 0.01	2.52 ^d,w^ ± 0.23	2.98 ^c,d,w,x^ ± 0.01	3.18 ^c,d,x^ ± 0.05
	Grapefruit peel powder	2.96 ^c,w^ ± 0.07	3.11 ^c,w^ ± 0.00	3.38 ^c,w^± 0.04	3.57 ^c,w^ ± 0.03
	Banana peel powder	3.45 ^b,w^ ± 0.01	3.98 ^b,x^ ± 0.01	4.34 ^b,x^ ± 0.25	4.90 ^b,y^ ± 0.06
Psychrotrophs					
	Control	3.13 ^a,w^ ± 0.02	3.60 ^a,w^ ± 0.20	4.10 ^a,x^ ± 0.20	4.97 ^a,y^ ± 0.21
	Lemon peel powder	<2.00 ^c,w^ ± 0.00	<2.00 ^c,w^ ± 0.00	<2.00 ^c,w^ ± 0.00	<2.00 ^c,w^ ± 0.00
	Orange peel powder	<2.00 ^c,w^ ± 0.00	<2.00 ^c,w^ ± 0.00	<2.00 ^c,w^ ± 0.00	<2.00 ^c,w^ ± 0.00
	Grapefruit peel powder	<2.00 ^c,w^ ± 0.00	<2.00 ^c,w^ ± 0.00	<2.00 ^c,w^ ± 0.00	<2.00 ^c,w^ ± 0.00
	Banana peel powder	<2.17 ^b,w^ ± 0.04	<2.39 ^b,w^ ± 0.11	2.46 ^b,w^ ± 0.00	2.60 ^b,w^ ± 0.13
*S. aureus*					
	Control	2.22 ^a,w^ ± 0.07	2.70 ^a,w,x^ ± 0.05	3.30 ^a,x,y^ ± 0.10	3.82 ^a,y^ ± 0.04
	Lemon peel powder	<2.00 ^b,w^ ± 0.00	<2.00 ^b,w^ ± 0.00	<2.00 ^c,w^ ± 0.00	<2.00 ^c,w^ ± 0.00
	Orange peel powder	<2.00 ^b,w^ ± 0.00	<2.00 ^b,w^ ± 0.00	<2.00 ^c,w^ ± 0.00	<2.00 ^c,w^ ± 0.00
	Grapefruit peel powder	<2.00 ^c,w^ ± 0.00	<2.00 ^c,w^ ± 0.00	<2.00 ^b,w^ ± 0.00	<2.00 ^b,w^ ± 0.00
	Banana peel powder	<2.00 ^b,w^ ± 0.00	<2.00 ^b,w^ ± 0.00	2.72 ^b,x^ ± 0.02	2.90 ^b,x^ ± 0.10
*Enterobacteriaceae*					
	Control	2.15 ^a,w^ ± 0.32	3.24 ^a,x^± 0.20	3.50 ^a,x^± 0.18	4.53 ^a,y^ ± 0.41
	Lemon peel powder	<2.00 ^b,w^ ± 0.00	<2.00 ^b,w^ ± 0.00	<2.00 ^c,w^ ± 0.00	<2.00 ^c,w^ ± 0.00
	Orange peel powder	<2.00 ^b,w^ ± 0.00	<2.00 ^b,w^ ± 0.00	<2.00 ^c,w^ ± 0.00	<2.00 ^c,w^ ± 0.00
	Grapefruit peel powder	<2.00 ^b,w^ ± 0.00	<2.00 ^b,w^ ± 0.00	<2.00 ^c,w^ ± 0.00	<2.00 ^c,w^ ± 0.00
	Banana peel powder	<2.00 ^b,w^ ± 0.00	<2.00 ^b,w,x^ ± 0.00	2.00 ^b,x^ ± 0.02	2.50 ^b,x^ ± 0.3

^a–d^ Mean values with different superscripts in the same column are significantly (*p* < 0.05) different. ^w–y^ Mean values with different superscripts in the same row are significantly (*p* < 0.05) different. The presented values constitute the mean of triplicate measurements ± standard error (SE).

**Table 3 foods-11-00301-t003:** Deterioration criteria of chicken patties with the incorporation of different fruit peel powders (1%) during frozen storage (−18 °C) for 3 months.

Parameters	Treatments	Storage Period (Months)			
		Time Zero	First Month	Second Month	Third Month
pH					
	Control	6.28 ^a,w^ ± 0.01	6.30 ^a,w,x^ ± 0.00	6.33 ^a,x^ ± 0.00	6.38 ^a,y^ ± 0.00
	Lemon peel powder	5.99 ^c,w^ ± 0.08	6.00 ^c,w^ ± 0.11	6.05 ^c,w^ ± 0.15	6.13 ^c,x^ ± 0.05
	Orange peel powder	6.15 ^b,w^ ± 0.00	6.16 ^b,w^ ± 0.00	6.19 ^b,w^ ± 0.01	6.24 ^b,x^ ± 0.01
	Grapefruit peel powder	6.15 ^b,w^ ± 0.01	6.18 ^b,w^ ± 0.03	6.24 ^b,x^ ± 0.01	6.26 ^b,x^ ± 0.01
	Banana peel powder	6.28 ^a,w^ ± 0.11	6.29 ^a,w^ ± 0.01	6.31 ^a,w^ ± 0.00	6.33 ^a,w^ ± 0.15
TBA (mg/kg)					
	Control	0.24 ^a,w^ ± 0.05	0.58 ^a,x^ ± 0.06	0.81 ^a,y^ ± 0.04	0.99 ^a,z^ ± 0.06
	Lemon peel powder	0.05 ^b,w^ ± 0.01	0.06 ^c,w^ ± 0.00	0.17 ^d,x^ ± 0.01	0.30 ^d,y^ ± 0.02
	Orange peel powder	0.05 ^b,w^ ± 0.01	0.09 ^c,w^ ± 0.00	0.22 ^d,x^ ± 0.01	0.37 ^c,d,y^ ± 0.01
	Grapefruit peel powder	0.08 ^b,w^ ± 0.02	0.11 ^c,w^ ± 0.04	0.30 ^c,x^ ± 0.02	0.42 ^c,y^ ± 0.03
	Banana peel powder	0.09 ^b,w^ ± 0.00	0.24 ^b,x^ ± 0.11	0.41 ^b,x^ ± 0.17	0.66 ^b,y^ ± 0.02
TVBN (mg/100 g)					
	Control	10.59 ^a,w^ ± 0.04	10.91 ^a,w^ ± 0.07	11.79 ^a,x^ ± 0.05	12.85 ^a,y^ ± 0.01
	Lemon peel powder	9.25 ^c,w^ ± 0.01	9.29 ^c,w^ ± 0.03	10.44 ^c,x^ ± 0.02	10.53 ^c,x^ ± 0.09
	Orange peel powder	9.41 ^c,w^ ± 0.04	9.45 ^c,w^ ± 0.01	10.55 ^c,x^ ± 0.05	10.58 ^c,x^ ± 0.05
	Grapefruit peel powder	9.53 ^c,w^ ± 0.02	9.56 ^c,w^ ± 0.02	10.62 ^c,x^ ± 0.06	10.67 ^c,x^ ± 0.00
	Banana peel powder	10.00 ^b,w^ ± 0.01	10.13 ^b,w^ ± 0.07	11.17 ^b,x^ ± 0.02	11.21 ^b,x^ ± 0.08

^a–d^ Mean values with different superscripts in the same column are significantly (*p* < 0.05) different. ^w–z^ Mean values with different superscripts in the same row are significantly (*p* < 0.05) different. TVBN—total volatile base nitrogen (mg/100 g); TBA—thiobarbituric acid (mg MAD/kg). The presented values constitute the mean of triplicate measurements ± standard error (SE).

**Table 4 foods-11-00301-t004:** Color values of chicken patties with the incorporation of different fruit peel powders (1%) during frozen storage (−18 °C) for 3 months.

Parameters	Treatments	Storage Period (Months)			
		Time Zero	First Month	Second Month	Third Month
L*					
	Control	57.71 ^a,w^ ± 0.23	60.00 ^a,x^ ± 0.10	62.07 ^a,y^ ± 0.14	65.23 ^a,z^ ± 0.50
	Lemon peel powder	53.12 ^b,w^ ± 0.20	54.30 ^b,w,x^ ± 0.15	56.22 ^b,x,y^ ± 0.20	59.09 ^b,y^ ± 0.19
	Orange peel powder	53.03 ^b,w^ ± 0.14	55.00 ^b,w,x^ ± 0.14	57.89 ^b,x,y^ ± 0.17	60.30 ^b,y^ ± 0.17
	Grapefruit peel powder	53.14 ^b.w^ ± 0.15	55.10 ^b,w,x^ ± 0.13	58.10 ^b,x,y^ ± 0.15	60.68 ^b,y^ ± 0.11
	Banana peel powder	47.07 ^c,w^ ± 0.27	49.25 ^c,w,x^ ± 0.11	49.18 ^c,w,x^ ± 0.32	50.10 ^c,x^ ± 0.22
a*					
	Control	11.31 ^b,w^ ± 0.15	10.00 ^b,x^ ± 0.10	7.89 ^b,y^ ± 0.63	6.00 ^b,z^ ± 0.32
	Lemon peel powder	12.76 ^a,w^ ± 0.16	11.40 ^a,x^ ± 0.15	9.37 ^a,y^ ± 0.11	9.20 ^a,y^ ± 0.16
	Orange peel powder	12.33 ^a,w^ ± 0.04	11.30 ^a,x^ ± 0.15	9.24 ^a,y^ ± 0.11	9.05 ^a,y^ ± 0.12
	Grapefruit peel powder	12.25 ^a,w^ ± 0.11	11.10 ^a,x^ ± 0.15	9.03 ^a,y^ ± 0.17	9.00 ^a,y^ ± 0.10
	Banana peel powder	11.40 ^b,w^ ± 0.22	10.50 ^b,w^ ± 0.15	8.00 ^b,x^ ± 0.26	7.11 ^b,x^ ± 0.10
b*					
	Control	21.54 ^c,w^ ± 0.09	21.10 ^c,w^ ± 0.13	20.73 ^c,w^ ± 0.11	18.75 ^c,x^ ± 0.08
	Lemon peel powder	23.40 ^b,w^ ± 0.24	23.20 ^b,w^ ± 0.17	23.05 ^b,w^ ± 0.11	23.26 ^b,w^ ± 0.11
	Orange peel powder	30.17 ^a,w^ ± 0.46	29.00 ^a,w^ ± 0.14	29.63 ^a,w^ ± 0.15	28.70 ^a,w^ ± 0.13
	Grapefruit peel powder	23.92 ^b,w^ ± 0.03	23.68 ^b,w^ ± 0.11	23.95 ^b,w^ ± 0.14	22.65 ^b,w^ ± 0.42
	Banana peel powder	14.17 ^d,w^ ± 0.10	13.50 ^d,w^ ± 0.13	13.56 ^d,w^ ± 0.15	13.35 ^d,w^ ± 0.11

^a–d^ Mean values with different superscripts in the same column are significantly (*p* < 0.05) different. ^w–z^ Mean values with different superscripts in the same row are significantly (*p* < 0.05) different. L*—lightness; a*—redness; b*—yellowness.; The presented values constitute the mean of triplicate measurements ± standard error (SE).

## Data Availability

All data generated or analyzed during this study are included in the submitted version of the manuscript.

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
