# Peer review of "Antioxidant and Antibacterial Effect of Fruit Peel Powders in Chicken Patties"

_foods, 2022, doi:10.3390/foods11030301_

Round 1

Reviewer 1 Report

Foods-1558056          

Antioxidant and antibacterial effect of fruit peel powders in 2 chicken patties

This study was carried out to investigate the efficiency of lemon, orange, grapefruit, and banana peel powders in improving the microbial quality, antioxidant activity, sensory attributes. One of the important studies also aiding in waste to product.

English is acceptable,

Figures and Tables are adequate and reasonable.

The addition of 1% of orange, lemon, grapefruit, or banana peel powders into chicken patties could improve their microbial quality, oxidative stability, and sensory attributes, and such improvements were more pronounced in lemon peel powder-treated samples.

With Regards,

Author Response

Reviewer #1:

COMMENT: This study was carried out to investigate the efficiency of lemon, orange, grapefruit, and banana peel powders in improving the microbial quality, antioxidant activity, sensory attributes. One of the important studies also aiding in waste to product. English is acceptable. Figures and Tables are adequate and reasonable. The addition of 1% of orange, lemon, grapefruit, or banana peel powders into chicken patties could improve their microbial quality, oxidative stability, and sensory attributes, and such improvements were more pronounced in lemon peel powder-treated samples.

With Regards,

ANSWER: Thank you very much for your time to review our manuscript, and for your positive comments and appreciations! We are delighted to read these lines! Thank you so much!

THANK YOU AGAIN FOR YOUR REVIEW!

Reviewer 2 Report

The present manuscript is an interesting reaserch. 

However, few suggestions are given to improve the draft further:

The abstract is too general and do not give any clue about the content of the paper. The authors need to specify these portion . It is not clear if the authors have targeted a specific period or they are talking generally. If they are talking generally, then there should be more literature added. On the other hand they are mentioning some names in key words, which are not touched in the abstract or title.

It was not clear enough what is the "prospective" that the authors seem to discuss in this topic. Moreover, the conclusion part must be rewrite. The authors should provide a concise and understandable conclusion of the work.

Author Response

COMMENT 1: The present manuscript is an interesting reaserch. However, few suggestions are given to improve the draft further.

ANSWER 1: Thank you very much for your time to review our manuscript, and for your positive comments and appreciations!

COMMENT 2: The abstract is too general and do not give any clue about the content of the paper. The authors need to specify these portions. It is not clear if the authors have targeted a specific period or they are talking generally. If they are talking generally, then there should be more literature added. On the other hand, they are mentioning some names in key words, which are not touched in the abstract or title.

ANSWER 2: The authors agree the reviewer recommendation! In this regard, new information consisting in the inclusion of the most important study results of the were inserted in the Abstract section.

The study was conducted during a period of three months, and each of the obtained results during the study monitoring was discussed in accordance with the available published data by other research team. All of the mentioned keywords are touched in the present new version of the Abstract section namely, “Chicken patties” and “physicochemical” – line 24; “Fruit peel powders” – line 23; “Phenolics” „Flavonoids” – line 25;

Thank you for your understanding!

COMMENT 3: It was not clear enough what is the "prospective" that the authors seem to discuss in this topic. Moreover, the conclusion part must be rewrite. The authors should provide a concise and understandable conclusion of the work.

ANSWER 3: The authors completely agree the reviewer comment. In this regard, the conclusion section was largely rewritten and new information were incorporated (see the lines 542 – 556 of the revised version).

THANK YOU AGAIN FOR YOUR REVIEW!

Reviewer 3 Report

Authors contributions:

In this paper, the authors have studied the effect of four different fruit peel powders: lemon, orange, grapefruit, and banana. Each of them is added in quantity of 1%.

The oxidative stability, microbial quality, physicochemical properties, and sensory attributes of chicken patties during 3 months of storage are studied at -18°C.

The total phenolics, flavonoids, and the antioxidant activity of fruit peel powders were analyzed.

According to the authors, lemon peel powder revealed the highest bioactive substance contents and free radical scavenging activity. Also, the fruit peel powders induced an increase in protein and decrease in fat contents with significant antioxidant and antibacterial activities.

In this study, the authors have found that sensory attributes are improved in lemon peel powder-treated patties. Also, total phenolic and total flavonoid contents together with DPPH% activity are significantly higher in lemon peel powder.

I have some reviewer notes:

In “Abstract” part. You have to give values of your most important results. Not only text description.

Section “2.5.7. Color evaluation of chicken patties”. How the colorimeter was calibrated? You have to describe the calibration procedure. (it is standard for Konica Minolta, model CR 410). Example: Prior to the start of the measurements, the instrument was pre-calibrated with a white CR-A43 plate (with illuminance characteristics D65: Y = 84.4; x = 0.3200 and y = 0.3365).

Section “2.6. Statistical analysis”. You have to describe the manufacturer and country of origin for SPSS statistics 27.0 software.

Figure 1 do not have horizontal axis title.

You have to add a description that the values are mean±SD. Also, Tables 2, 3, 4.

The “Discussion” is well presented. For every result and as a summary.

You have to improve your “Conclusion” part. You have to describe: What are your results? How your results improve the known solution in this study area? What new knowledge you give with your results? What is the weak part of your research? How your research will be continued?

I have some suggestions:

Improve the way of presentation of your results.

In your next papers you can calculate some of the color indices. White, yellow, browning indices. They can give you more information about the change of color than simple using of L*, a* and b* color components.

You can check this paper:

https://www.researchgate.net/publication/354707117_RESEARCH_ON_COLOR_DATA_FROM_ELEMENTS_OF_FOLK_COSTUMES

about the color indices. They are described under Figure 1.

I suggest the color indices only for your next researches.

Author Response

Authors contributions:

COMMENT 1: In this paper, the authors have studied the effect of four different fruit peel powders: lemon, orange, grapefruit, and banana. Each of them is added in quantity of 1%.The oxidative stability, microbial quality, physicochemical properties, and sensory attributes of chicken patties during 3 months of storage are studied at -18°C. The total phenolics, flavonoids, and the antioxidant activity of fruit peel powders were analyzed. According to the authors, lemon peel powder revealed the highest bioactive substance contents and free radical scavenging activity. Also, the fruit peel powders induced an increase in protein and decrease in fat contents with significant antioxidant and antibacterial activities. In this study, the authors have found that sensory attributes are improved in lemon peel powder-treated patties. Also, total phenolic and total flavonoid contents together with DPPH% activity are significantly higher in lemon peel powder.

ANSWER 1: Thank you very much for your time to review our manuscript, and for your positive comments and appreciations! We are delighted to read these lines! Thank you so much!

I have some reviewer notes:

COMMENT 2: In “Abstract” part. You have to give values of your most important results. Not only text description.

ANSWER 2: The authors completely agree the reviewer suggestion. As requested, the most important values were inserted in the Abstract section, in order to reflect and emphasize the most important findings. Please see the lines 27-30 of the revised version.

COMMENT 3: Section “2.5.7. Color evaluation of chicken patties”. How the colorimeter was calibrated? You have to describe the calibration procedure. (it is standard for Konica Minolta, model CR 410). Example: Prior to the start of the measurements, the instrument was pre-calibrated with a white CR-A43 plate (with illuminance characteristics D65: Y = 84.4; x = 0.3200 and y = 0.3365).

ANSWER 3: According to the reviewer requirement, the following sentences were inserted in the revised version of the manuscript (see lines 209-215):

The color of chicken patties from different groups was measured after processing, and monthly during frozen storage, using a Chroma Meter (Konica Minolta, model CR 410, Japan) with illuminant D65, 11 mm aperture size, and 10° observer, according to the method described by Shin et al. [20]. The meter was calibrated against a white CR410 calibration plate (Y=94.40, x=0.3159, y=0.3325). From each sample, three reading were taken and the mean value was calculated and expressed as CIE lightness (L*), redness (a*), and yellowness (b*).

COMMENT 4: Section “2.6. Statistical analysis”. You have to describe the manufacturer and country of origin for SPSS statistics 27.0 software.

ANSWER 4: According to the reviewer requirement, the following sentences were inserted in the revised version of the manuscript (see lines 233-234 of the revised version): „... IBM SPSS statistics for Windows, version 27.0 (IBM Corp., Armonk, NY, USA).”

COMMENT 5: Figure 1 do not have horizontal axis title.

ANSWER 5: Thank you for the suggestion. „Different fruit peel powders” – was inserted as definition for the horizontal axis

COMMENT 6: You have to add a description that the values are mean±SD. Also, Tables 2, 3, 4.

ANSWER 6: Thank you for the suggestion. This sentence „The presented values contitute the mean of triplicate measurements ± SE.” was inserted under the figure 1 and each table.

COMMENT 7: The “Discussion” is well presented. For every result and as a summary.

ANSWER 7: Thank you for your appreciations!

COMMENT 8: You have to improve your “Conclusion” part. You have to describe: What are your results? How your results improve the known solution in this study area? What new knowledge you give with your results? What is the weak part of your research? How your research will be continued?

ANSWER 8: Thank you for your comments. The requested information were incorporated in the new revised version of the manuscript. See the lines 542-556.

COMMENT 9: I have some suggestions:

Improve the way of presentation of your results. In your next papers you can calculate some of the color indices. White, yellow, browning indices. They can give you more information about the change of color than simple using of L*, a* and b* color components.You can check this paper:

https://www.researchgate.net/publication/354707117_RESEARCH_ON_COLOR_DATA_FROM_ELEMENTS_OF_FOLK_COSTUMES

about the color indices. They are described under Figure 1.

I suggest the color indices only for your next researches.

ANSWER 9: Thank you very much! All of these valuable suggestions will be taken into consideration in our next researches.

THANK YOU AGAIN FOR YOUR REVIEW!

Round 2

Reviewer 3 Report

The paper is corrected according to the reviewer's requirements.

All necessary comments are included.

Author Response

Thank you again for your positive appreciations.